# Multi-Task Knowledge Distillation with Embedding Constraints for Scholarly Keyphrase Boundary Classification

**Seo Yeon Park and Cornelia Caragea**
Computer Science
University of Illinois Chicago
spark313@uic.edu    cornelia@uic.edu

## Abstract

The task of scholarly keyphrase boundary classification aims at identifying keyphrases from scientific papers and classifying them with their types from a set of predefined classes (e.g., task, process, or material). Despite the importance of keyphrases and their types in many downstream applications including indexing, searching, and question answering over scientific documents, scholarly keyphrase boundary classification is still an under-explored task. In this work, we propose a novel embedding constraint on multi-task knowledge distillation which enforces the teachers (single-task models) and the student (multi-task model) similarity in the embedding space. Specifically, we enforce that the student model is trained not only to imitate the teachers' output distribution over classes, but also to produce language representations that are similar to those produced by the teachers. Our results show that the proposed approach outperforms previous works and strong baselines on three datasets of scientific documents.

## 1 Introduction

Scholarly keyphrase boundary classification is the task of identifying highly summative phrases from scientific papers and classifying them into a set of pre-defined classes (Augenstein et al., 2017; Augenstein and Søgaard, 2017). In a scientific domain, keyphrases and their classes, e.g., task, process, or material, are critical for many downstream applications including effectively mining, searching, and analyzing the scientific literature (Augenstein et al., 2017); promoting an efficient understanding of what methods, processes, tasks, or resources are being used or proposed in a given paper and how to track them over time (Uban et al., 2021); scientific paper summarization (Abu-Jbara and Radev, 2011; Qazvinian et al., 2010); keyphrase-based question answering (Quarteroni and Manandhar, 2006); machine comprehension (Subramanian et al., 2018); scientific paper rec-

ommendation (Chen et al., 2015); topic classification (Sadat and Caragea, 2022; Onan et al., 2016; Caragea et al., 2015); and, more broadly, data augmentation for NLP tasks (Li et al., 2023a,b).

The scholarly keyphrase boundary classification (SKBC) task shares similarities with the named entity recognition (NER) task. Specifically, NER aims to identify entities' boundaries and classify them into pre-defined categories (e.g.,'U.N.' is an 'organization' entity) (Collobert et al., 2011). Similarly, SKBC aims to identify the boundary of keyphrases or entities in scientific text and classify them into a set of pre-defined classes (e.g., 'WebVision' is a 'dataset' entity and 'object detection' is a 'task' entity). NER (Tjong Kim Sang and De Meulder, 2003; Ratinov and Roth, 2009; Collobert et al., 2011; Lample et al., 2016) is treated as a fundamental task in NLP because the named entities often convey the key information of a text. Similar to NER, the key information of the scientific discourse available in research papers revolves around scientific entities (e.g., datasets, tasks, models). Despite that much research has been done on NER (Bodapati et al., 2019; Yu et al., 2020; Chen et al., 2020; Tabassum et al., 2020; Jia and Zhang, 2020; Li et al., 2020; Yan et al., 2021; Zhou et al., 2022; Zhang et al., 2022; Liu and Ritter, 2023), there are only very few works that exist on SKBC (Augenstein and Søgaard, 2017; Luan et al., 2017, 2018; Park and Caragea, 2020; Jain et al., 2020), and hence, this task of scholarly keyphrase boundary classification is very much under-explored. One potential reason for this under-exploration is the lack of large annotated datasets for this task and the difficulty of annotations which require human experts in a scientific field.

In this paper, we propose a novel framework for SKBC, called multi-task knowledge distillation with cosine embedding constraints. Clark et al. (2019b) showed the effectiveness of the multi-task knowledge distillation framework which dis-

tills knowledge from multiple teachers (single-task models) to the student model (multi-task model). However, in the process of learning from multiple teachers corresponding to multiple related auxiliary tasks, the student model may become biased towards a particular auxiliary task (or a subset of these tasks) and may "erase" all the "good" information (linguistic features that "matter") from the other remaining tasks. To overcome this, we propose a way of imposing similarity constraints in the embedding space between the student and all the teachers to ensure that the student's final representations are not biased towards some auxiliary task(s), and hence, the student's representations do not diverge (too much) from those of any of the teachers. Consequently, our proposed novel embedding constraint lets the student model be focused on the target task by optimally using the knowledge learned from all auxiliary tasks. Notably, we achieve state-of-the-art performance on three datasets with our proposed multi-task knowledge distillation with embedding constraints. To facilitate future research on scholarly keyphrase classification, we release our code.[1]

Our contributions are summarized as follows:

- We propose a novel cosine embedding constraint added to multi-task knowledge distillation for keyphrase boundary classification that enforces teacher-student similarity in the embedding space.

- We conduct comprehensive evaluation on three datasets. Experiments show that our proposed approach significantly outperforms strong baselines and prior works and achieves new state-of-the-art on this task.

- We provide a detailed analysis using entropy to understand our proposed model's behavior (uncertainty) in predicting keyphrases, together with an error analysis.

## 2 Related Work

The task of automatically identifying and classifying keyphrases from scientific papers has long been under-explored but has recently started to gain attention in the research community. For example, QasemiZadeh and Schumann (2016) motivated the importance of this task for many applications such as knowledge acquisition and topic tracking, and

were among the first to create a dataset annotated for terminology extraction and classification from ACL Anthology papers to stir research in this area. Augenstein et al. (2017) proposed a shared task on ScienceIE at SemEval 2017, which includes keyphrase identification and keyphrase classification as Subtask A and Subtask B, respectively. This shared task attracted a large number of participating teams and achieved a highest F1-score of $56\%$ and $44\%$ for Subtasks A and B, respectively, using an RNN with a CRF layer on top (Augenstein et al., 2017). These low F1 scores proved the difficulty of the subtasks. Other works participating in this shared task used hand-crafted features for the keyphrase identification and classification. For example, Liu et al. (2017) used combined features of pretrained word embeddings and a set of linguistic features with Support Vector Machines. Lee et al. (2017) and Marsi et al. (2017) used CRF with linguistically motivated features such as Part-of-Speech tags and word lemmas.

One observation that was made apparent was that the existing annotated datasets for this task are small in size. In order to overcome the small data size problem, Augenstein and Søgaard (2017) proposed to transfer knowledge from data-rich tasks through deep multi-task learning where keyphrase boudary classification represents the main task and several related tasks such as FrameNet, semantic supersense tagging, chunking, and multiword expression identification serve as auxiliary tasks (one at a time) to help guide the main task. Still, the results of this approach are low (Augenstein and Søgaard, 2017), e.g., yielding an F1 of $45.49\%$ on the SemEval 2017 dataset (Augenstein et al., 2017), and an F1 of $58.95\%$ on the ACL dataset (QasemiZadeh and Schumann, 2016) for keyphrase classification. Park and Caragea (2020) proposed to use intermediate task transfer learning using pre-trained language models (Pruksachatkun et al., 2020) to overcome the data scarcity problem by using one intermediate task at a time. In another line of research, Luan et al. (2017) aimed at addressing the same small data size problem and proposed a graph-based semi-supervised algorithm with a data selection strategy to leverage unannotated articles. Lai et al. (2020) introduced a semi-supervised approach for scholarly keyphrase identification employing self-knowledge distillation. Ammar et al. (2017) also used semi-supervised learning for entity and relation extraction from scientific papers.

---

[1] https://github.com/seoyeon-p/MTL-KD-SKIC

| Target Input | | Nuclear | theory | devoted | major | efforts | to | describe | thermalization |
|---|---|---|---|---|---|---|---|---|---|
| Target Output | Identification | B | I | O | O | O | O | O | B |
| | Classification | B-Task | I-Task | O | O | O | O | O | B-Process |

Table 1: An example instance of scholarly keyphrase boundary classification with BIO schema.

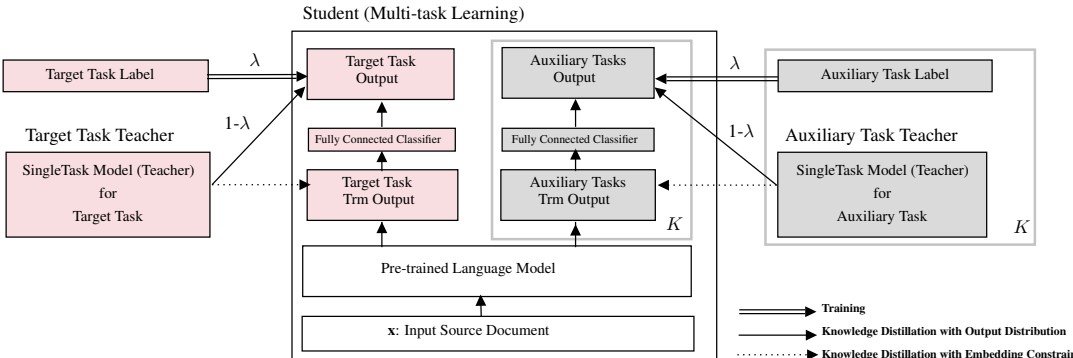

Figure 1: An overview of our proposed approach using plate notation. Plates (the grey boxes in the figure indexed by $K$) refer to repetitions of knowledge distilling steps for $K$ auxiliary tasks. The single task model is used as a teacher model for our multi-task student model. "Trm Output" refers to the last layer of pre-trained language models.

Other works used supervised or semi-supervised keyphrase extraction or generation (Kulkarni et al., 2022; Patel and Caragea, 2021, 2019; Alzaidy et al., 2019; Florescu and Caragea, 2017; Ye and Wang, 2018; Chowdhury et al., 2022; Garg et al., 2023, 2022; Ye et al., 2021; Huang et al., 2021; Wu et al., 2023). Moreover, Ammar et al. (2018) focused on building a literature (heterogeneous) graph to organize published scientific literature for easy manipulation and discovery. Luan et al. (2018) created a new dataset annotated for three tasks, sentence-level scientific entity recognition, relation extraction, and coreference resolution, and used a multi-task learning framework (SciIE) to jointly learn from all three tasks. Jain et al. (2020) extended sentence-level information extraction to document-level, creating a comprehensive dataset for extracting information from scientific articles. Pan et al. (2023) introduced a scholarly benchmark for dataset mention detection to address the limitations of existing corpora in terms of size, diversity of dataset mentions, and entity linking information. In contrast, we propose a cosine embedding constraint added to multi-task knowledge distillation that enforces teacher-student similarity in the embedding space. Similar to our work, some works (Sun et al., 2019; Aguilar et al., 2020) explored teacher-student similarity at the internal representations. However, their focus is on model compression (to approximate the teacher with a smaller student model) while our work is focused on teacher-student semantic space similarity that

ensures the student's final representations do not diverge (too much) from those of the teachers. Accordingly, we aim to preserve the beneficial information from the teachers that may not be retained in the multi-task knowledge distillation due to catastrophic forgetting (Kirkpatrick et al., 2016). Along similar lines, Mahmoud and Hajj (2022) presented multi-objective learning which leveraged knowledge distillation in a multi-task autoencoder network to minimize catastrophic forgetting of pre-trained tasks. Meshgi et al. (2022) introduced a regularization term in multi-task learning, utilizing task uncertainty to enhance model generalization and mitigate the risk of catastrophic forgetting.

## 3 Target Task

**Problem Definition** The task objective is to identify and classify keyphrase boundaries from an input text sequence. Let $\mathbf{x} = (x_1, \cdots, x_n)$ be an input sequence of size $n$, where $x_i, i = 1, \cdots, n$ are words from a vocabulary $V$, and let $L$ be a set of labels. The goal is to learn a function (or a classifier) $f : (x_i) \rightarrow y_i$ that classifies each word $x_i$ from $\mathbf{x}$ into one of the available labels $y_i \in L$. We do this by minimizing the cross-entropy loss. The set $L$ follows the BIO schema for keyphrase identification, where 'B' refers to the beginning of a keyphrase, 'I' to the inside of a keyphrase, 'O' is any word that is not part of a keyphrase (we used a similar labeling scheme for keyphrase classification, e.g., 'B-Task', 'I-Task', 'B-Process', 'I-Process', 'O'). Table 1 shows an input-output sequence pair of both keyphrase identification and classification.

## 4 Approach

In this section, we first describe our baseline multi-task learning model (§4.1). We then present multi-task learning that distills knowledge from multiple related auxiliary single-tasks (teachers) into a multi-task (student) model (§4.2). After that, we describe our multi-task knowledge distillation with embedding constraints (§4.3). Last, we describe the auxiliary tasks (§4.4).

### 4.1 Multi-Task Learning

In multi-task learning (MTL) a target task is learned by leveraging knowledge from multiple related auxiliary tasks (Liu et al., 2019b).

Our baseline MTL model is similar to that of Liu et al. (2019b) and is shown in Figure 1 (middle part). All our models are built on top of pre-trained language models, namely BERT (Devlin et al., 2019) and SciBERT (Beltagy et al., 2019).

As shown in the figure, the lower layers (corresponding to a pre-trained language model $\mathcal{M}$) are shared across all tasks, whereas the top layers are task-specific. The model jointly learns the target task with different types of NLU tasks—natural language inference tasks, single-sentence classification, pairwise text classification, and sequence tagging tasks. For the target task, the model $\mathcal{M}$ yields a sequence of contextual embeddings, one for each token $x_i$ in the input sequence $\mathbf{x}$ as $\mathbf{h}_i = \mathcal{M}(x_i) \in \mathbb{R}^d$ (of dimension $d$). The obtained representation of each token is then sent through a fully connected layer with softmax for classification, i.e., $\hat{y}_i = softmax(\mathbf{W}\mathbf{h}_i)$, where $\mathbf{W} \in \mathbb{R}^{d \times |L|}$ is a weight matrix to be learned, and $\mathbf{h}_i$ is the contextual embedding of the $i^{th}$ token $x_i$ in the input sequence $\mathbf{x}$. For the NLU classification tasks, we use the [CLS] token for classification. In MTL, we train the model to minimize the sum of cross-entropy loss for all tasks. Unlike previous work (Augenstein and Søgaard, 2017), we use all auxiliary tasks at once in our MTL model.

### 4.2 Multi-Task Learning with Knowledge Distillation

In standard knowledge distillation (KD) (Hinton et al., 2015), a student model is trained to imitate a teacher model's output distribution under the assumption that this output distribution provides better signal to the student than the gold label itself. Clark et al. (2019b) showed that using knowledge distillation with MTL when learning from multiple

related tasks can improve the performance of models that use MTL with only one related task. We describe MTL+KD (Clark et al., 2019b) as follows:

Given a set of $K + 1$ tasks, i.e., a target task and $K$ auxiliary tasks (see Figure 1), each with its own training set $\mathcal{D}^k$, $k = 1, \cdots, K + 1$, we train a single-task (teacher) model on each task, denoted $\theta^k$. Then we use the single-task models $\theta^k$ to teach a multi-task shared (student) model with parameters $\theta$ using cross-entropy loss $\mathcal{L}_{CE}$ given as follows:

$$\mathcal{L}_{CE}(\theta) = \sum_{k=1}^{K+1} \sum_{(x_i^k, y_i^k) \in \mathcal{D}^k} \ell(\ f^k(x_i^k, \theta^k)\ ,\ f^k(x_i^k, \theta)\ ) \quad (1)$$

where $f^k(x_i^k, *)$ denotes the output for task $k$ produced by a neural model with parameters $*$. However, Clark et al. (2019b) pointed out that there is a limitation that the student might not be able to overcome the teacher model's performance and suggested to use teacher annealing KD, which combines gold labels with predictions. Thus, the cross-entropy loss with teacher annealing becomes:

$$\mathcal{L}_{CE}(\theta) = \sum_{k=1}^{K+1} \sum_{(x_i^k, y_i^k) \in \mathcal{D}^k} \ell(\ \lambda y_i^k + (1 - \lambda)f^k(x_i^k, \theta^k)\ ,$$
$$f^k(x_i^k, \theta)\ ) \quad (2)$$

where $\lambda$ is linearly increased from 0 to 1 through training (see Figure 1).

### 4.3 Multi-Task Knowledge Distillation with Embedding Constraints

We now present our multi-task knowledge distillation with embedding constraints framework which enforces "teacher-student" similarity in the embedding space in addition to the student imitating the teachers' output distribution. That is, we constrain the multi-task student model to learn language representations that are similar in the semantic space with those of the teachers.

Despite that large pre-trained language models (Devlin et al., 2019) fine-tuned on supervised tasks perform remarkably well due to their capability to capture the structure of language (Clark et al., 2019a) and other linguistics patterns, e.g., aspects of syntax (Shi et al., 2016; Blevins et al., 2018), in

| Task Name | \|Train\| | \|Dev\| | \|Class\| | Task Type | Domain |
|---|---|---|---|---|---|
| Supersense | 8,006 | 1,186 | 3 | Sequence tagging | Newswire, Novel |
| MWEs | 11,890 | 1,146 | 57 | Sequence tagging | Online review |
| FrameNet | 9,334 | 4,000 | 92 | Sequence tagging | Open domain |
| Chunking | 104,054 | 11,588 | 12 | Sequence tagging | Wall Street Journal |
| POS Tagging (POS) | 334,180 | 54,138 | 15 | Sequence tagging | Wikipedia, Talks, Literature |
| NER | 13.996 | 3,241 | 12 | Sequence tagging | Wall Street Journal |
| SciTail | 23,596 | 1,304 | 2 | Natural language inference | Science exams |
| WNLI | 634 | 146 | 2 | Natural language inference | Fiction books |
| RTE | 2,490 | 277 | 2 | Natural language inference | News, Wikipedia |
| MRPC | 3,688 | 1,725 | 2 | Paraphrase | News |
| CoLA | 8,551 | 1,043 | 2 | Acceptability | Books, Journal articles |

Table 2: The overview of auxiliary tasks in our experiments.

| Task Name | \|Train\| | \|Dev\| | \|Test\| | \|Class\| | \|Avg KP per Doc\| | \|Avg Doc Word Count\| | Domain |
|---|---|---|---|---|---|---|---|
| SemEval 2017 | 5,992 | 1,076 | 1,817 | 3 | 18 | 162 | Computer Science, Physics, and Material Science |
| ACL RD-TEC 2.0 | 1,930 | 214 | 1,088 | 7 | 12 | 107 | Natural Language Processing |
| SciIE | 5,712 | 641 | 1,677 | 6 | 17 | 120 | Artificial Intelligence |

Table 3: The statistics of target tasks in our experiments.

MTL+KD, some information from the pre-trained teachers $\mathcal{M}$ may not be retained in the MTL model possibly due to catastrophic forgetting (Kirkpatrick et al., 2016). To preserve the "good" information from the teachers, e.g., linguistic features that "matter," we constrain the model to enforce the similarity between the word embeddings produced by the student with those produced by the teachers.

We calculate the similarity between the hidden representations (i.e., the last transformer layer outputs) $\mathbf{h}_i^\theta$ produced by the student with those produced by the teachers $\mathbf{h}_i^{\theta^k}$ via cosine similarity. Thus, we define an additional loss term to penalize the student model from diverging from the hidden representations of the teachers:

$$\mathcal{L}_{cos}(\theta) = \sum_{k=1}^{K+1} \sum_{(x_i^k, y_i^k) \in \mathcal{D}^k} 1 - cos(\mathbf{h}_i^{\theta^k}, \mathbf{h}_i^\theta)$$

(3)

where $\mathbf{h}_i^{\theta^k} = \mathcal{M}(x_i^k) \in \mathbb{R}^d$ is the hidden representation for sample $x_i^k$ of task $k$ produced by the corresponding teacher $\mathcal{M}$ (similarly for $\mathbf{h}_i^\theta$). Our final loss consists of (i) cross-entropy loss from Eq. 2; and (ii) cosine embedding loss from Eq. 3 as:

$$\mathcal{L}(\theta) = \mathcal{L}_{CE}(\theta) + \alpha \mathcal{L}_{cos}(\theta),$$

(4)

where $\alpha$ is a hyper-parameter.

### 4.4 Auxiliary Tasks

Training on multiple tasks is known to help regularize multi-task models (Ruder, 2017). Here, we select $K = 11$ auxiliary tasks across a range of GLUE benchmarks (Wang et al., 2019), e.g.,

natural language inference, sequence tagging, paraphrase detection, and grammatical acceptability tasks. These tasks have proven beneficial in multi-task learning (Liu et al., 2019b). We show the statistics of our auxiliary tasks in Table 2. In detail, we use the following tasks: Supersense (Johannsen et al., 2014), MWEs (Schneider and Wooters, 2017), FrameNet (Das et al., 2014), Chunking (Tjong Kim Sang and Buchholz, 2000), POS Tagging (McDonald et al., 2013), NER (Tjong Kim Sang and De Meulder, 2003), SciTail (Khot et al., 2018), WNLI (Wang et al., 2019), RTE (Wang et al., 2019), MRPC (Dolan and Brockett, 2005), and CoLA (Warstadt et al., 2019). Further details about these tasks are presented in Appendix A.1.

### 5 Target Task Datasets

To evaluate models' performance on our target task, i.e., keyphrase boundary identification and classification, we use three datasets, as described below. Table 3 shows the details of each dataset.

**SemEval 2017 Task 10** SemEval 2017 (Augenstein et al., 2017) is a scientific keyphrase boundary identification and classification dataset. The dataset has three pre-defined classes (keyphrase types) which are *Process*, *Task*, and *Material*.

**ACL RD-TEC 2.0** ACL (QasemiZadeh and Schumann, 2016) is a dataset for term and entity categorization in scientific text. The ACL dataset has seven pre-defined classes, which are *Tool, Measurement, Language Resources (LR), Language Resources Product (Lr-prod), Model, Technology*

*(Tech),* and *OtherScientificTerm (Other).* Note that the class *Other* here is different from the class *Other* ('O') in the BIO scheme that represents non-keyphrases.

**SciIE**  SciIE (Luan et al., 2018) is a dataset for the detection of scientific entities, their relations, and coreference clusters. The SciIE dataset has six predefined classes, which are *Material, Method, Task, OtherScientificTerm (Other), Metric,* and *Generic.*

## 6  Experiments

### 6.1  Baselines

We use the following baselines for comparison.

- **BiLSTM** (Augenstein and Søgaard, 2017): Single target task learning of 3-layer BiLSTM with pretrained SENNA embeddings.[2]

- **BiLSTM + MTL + \*** (Augenstein and Søgaard, 2017): Multitask learning of 3-layer BiLSTMs with SENNA embeddings. Note that \* corresponds to one auxiliary task.

- **BERT** (Devlin et al., 2019) BERT fine-tuning on the target task.

- **BERT + ITTL** (Park and Caragea, 2020): BERT with intermediate task transfer learning which fine-tunes BERT on a single auxiliary task before fine-tuning on the target task.

- **Self-Distill** (Lai et al., 2020): A semi-supervised learning that trains a single teacher BERT model on labeled data and is used to generate pseudo-labels on unlabeled data. Afterward, it combines the labeled and pseudo-labeled data to train a single-task student BERT model that is initialized by the teacher BERT model.

- **SciBERT** (Beltagy et al., 2019): SciBERT fine-tuning on the target task.

- **SciBERT + ITTL** (Park and Caragea, 2020): SciBERT with intermediate task transfer learning which fine-tunes SciBERT on a single auxiliary task before fine-tuning SciBERT on the target task.

- **SciBERT-MKD** (Liu et al., 2019a): A multi-task knowledge distillation framework distilling knowledge from a multi-task teacher SciBERT to a multi-task student SciBERT, utilizing the same auxiliary tasks as our proposed method for training both teacher and student multi-task models.

[2]https://ronan.collobert.com/senna/

To compare our method with existing NER approaches, we consider the following two NER methods that utilize a backbone of BiLSTM and BERT, respectively.

- **Biaffine NER** (Yu et al., 2020): A framework that has a biaffine model on top of a multi-layer BiLSTM for a graph-based dependency parsing to provide a global view on the input for NER.

- **Unified MRC for NER** (Li et al., 2020): A unified framework that has BERT as a backbone which treats the task of NER as a machine reading comprehension (MRC) task.

### 6.2  Implementation

For the SemEval 2017 and SciIE, we use the published train, validation, and test splits. For the ACL RD-TEC 2.0 dataset, we perform 60/10/30 split to create the train, validation, and test splits, respectively. We estimate hyper-parameters for each deep neural model via a grid search over combinations. The detailed information on the best hyper-parameters setting is shown in Appendix A.2.

### 6.3  Main Results

We show the main experimental results in Table 4. We perform both keyphrase boundary identification (*Identif*) and keyphrase boundary classification (*Classif*). We evaluate the performance of models using phrase-level micro-averaged F1-score and make the following observations.

First, we observe that the SciBERT-based models generally achieve higher performance compared with the previous models based on BiLSTM especially on keyphrase classification. Furthermore, we observe that SciBERT-based models perform better than BERT-based models. For example, vanilla SciBERT achieves 66.70/48.21 F1 on SemEval keyphrase identification/classification, whereas vanilla BERT achieves 60.40/46.82 F1. We posit that the lower performance of BERT is in part due to distributional shifts (i.e., domain differences from the general to the scientific domain).

Second, we observe that SciBERT can be improved further by using multi-task knowledge distillation. Distilling knowledge from the teachers to the student by training the student to imitate the teachers' output probability for all tasks, i.e., the target and all auxiliary tasks (SciBERT+MTL+KD+ALL) improves the F1 score to 74.17/55.81 ($\Delta$ = 7.47/7.60 over SciBERT)

| | SemEval 2017 Task 10 | | ACL RD-TEC 2.0 | | SciIE | |
| --- | --- | --- | --- | --- | --- | --- |
| | *Identif* | *Classif* | *Identif* | *Classif* | *Identif* | *Classif* |
| *Results on BiLSTM-based Models* | | | | | | |
| BiLSTM (Augenstein and Søgaard, 2017) | 67.70 | 38.01 | 81.85 | 58.51 | 72.33 | 58.05 |
| BiLSTM + MTL + Supersense | 63.93 | 43.54 | 81.36 | 58.95 | 72.65 | 54.33 |
| BiLSTM + MTL + MWEs | 72.42 | 45.49 | 80.69 | 56.87 | 72.92 | 55.21 |
| BiLSTM + MTL + FrameNet | 65.18 | 45.24 | 81.68 | 58.89 | 70.44 | 52.60 |
| BiLSTM + MTL + Chunking | 63.96 | 42.86 | 81.37 | 57.84 | 75.40 | 59.43 |
| BiLSTM + MTL + ALL | 67.06 | 43.75 | 77.89 | 55.10 | 80.51 | 63.80 |
| Biaffine-NER (Yu et al., 2020) | 68.76 | 47.56 | 82.04 | 59.49 | 80.58 | 63.76 |
| *Results on BERT-based Models* | | | | | | |
| BERT (Devlin et al., 2019) | 60.40 | 46.82 | 79.50 | 51.67 | 81.02 | 65.44 |
| BERT + ITTL (Park and Caragea, 2020) | 65.89 | 49.02 | 77.72 | 45.77 | 75.87 | 57.40 |
| Self-Distill (Lai et al., 2020) | 55.40 | 41.56 | 76.66 | 50.38 | 73.57 | 56.95 |
| Unified MRC for NER (Li et al., 2020) | 65.29 | 50.05 | 81.88 | 55.36 | 85.41 | 68.81 |
| BERT + MTL + ALL | 62.73 | 47.18 | 79.92 | 56.64 | 83.99 | 66.72 |
| BERT + MTL + KD + Target | 67.91 | 51.95 | 83.33 | 63.65 | 87.27 | 69.44 |
| BERT + MTL + KD + Target + Cosine | 66.21 | 51.17 | 78.60 | 57.52 | 87.17 | 68.68 |
| BERT + MTL + KD + ALL | 65.81 | 49.44 | 80.77 | 55.55 | 85.88 | 67.54 |
| BERT + MTL + KD + ALL + Cosine | 66.47 | 49.67 | 78.45 | 54.60 | 87.49 | 68.18 |
| *Results on SciBERT-based Models* | | | | | | |
| SciBERT (Beltagy et al., 2019) | 66.70 | 48.21 | 79.77 | 65.11 | 81.02 | 67.44 |
| SciBERT + ITTL (Park and Caragea, 2020) | 73.89 | 56.90 | 88.01 | 69.90 | 82.65 | 75.23 |
| SciBERT-MKD (Liu et al., 2019a) | 73.69 | 54.82 | 83.34 | 68.85 | 81.39 | 70.06 |
| SciBERT + MTL + ALL | 69.04 | 52.30 | 85.71 | 68.85 | 87.30 | 72.99 |
| SciBERT + MTL + KD + Target | 70.47 | 53.38 | 87.07 | 70.18 | 88.72 | 73.61 |
| SciBERT + MTL + KD + Target + Cosine | 70.91 | 54.60 | 85.69 | 69.72 | 90.49 | 76.23 |
| SciBERT + MTL + KD + ALL | 74.17 | 55.81 | 88.37 | 71.33 | 89.58 | 76.18 |
| SciBERT + MTL + KD + ALL + Cosine | **75.08** | **57.08** | **89.62** | **73.09** | **90.87** | **77.30** |

Table 4: F1-score results of proposed models based on BERT and SciBERT in comparison with the previous work. For pre-trained language models with intermediate task transfer learning (ITTL), we use the best-reported results from prior work. Underlined scores are best within each group and bold scores are best overall.

on SemEval. Remarkably, the best result is achieved with our proposed model, which enforces teacher-student similarity in the embedding space. Precisely, our proposed model (SciBERT+MTL+KD+ALL+Cosine) improves the F1 score further to 75.08/57.08 ($\Delta = 8.38/8.87$ over SciBERT) on SemEval. While SciBERT with intermediate task transfer learning (SciBERT + ITTL) shows competitive performance compared with vanilla SciBERT, our proposed model outperforms SciBERT + ITTL on all three datasets. Our method also outperforms existing multi-task knowledge distillation baseline methods (i.e., Self-Distill and SciBERT-MKD) on all three datasets. For example, our method achieves a better F1 score of 75.08/57.08 on SemEval, compared to SciBERT-MKD which achieves an F1 score of 73.69/54.82 on keyphrase identification/classification. This proves the effectiveness of our proposed method. In addition, using knowledge distillation only on the target task, KD+Target, we notice a performance decrease from KD+ALL. Interestingly, in contrast

to SciBERT, BERT returns the best performance when distilling knowledge by training the student to imitate the teacher's output applied only for the target task. While BERT+MTL+KD+* models improve over vanilla BERT, there is no improvement when teacher-student similarity is enforced in the embedding space for BERT. A potential explanation is vocabulary shifts (i.e., domain differences with our target task) between BERT and our scientific domain datasets. Specifically, BERT vocabulary is from a general domain whereas the vocabulary of our target task is from a scientific domain. In contrast to BERT, SciBERT vocabulary is from a scientific domain. The token overlap between BERT vocabulary and SciBERT vocabulary is only 42%, which implies that there is a substantial difference in frequently used words between scientific and general domain texts (Beltagy et al., 2019). We conclude that SciBERT has more coverage in handling scholarly texts and results in better language representations for scholarly keyphrase classification than BERT.

| | SemEval 2017 Task 10 | | ACL RD-TEC 2.0 | | SciIE | |
|---|---|---|---|---|---|---|
| | *Identif* | *Classif* | *Identif* | *Classif* | *Identif* | *Classif* |
| **SciBERT + MTL + Chunking** | 72.66 | 55.17 | 86.27 | 71.87 | 89.28 | 76.76 |
| **SciBERT + MTL + Chunking + KD + Both** | 72.50 | 53.88 | 85.36 | 71.27 | 87.49 | 75.58 |
| **SciBERT + MTL + Chunking + KD + Both + Cosine** | 72.12 | 54.93 | 87.92 | 70.23 | 86.93 | 71.46 |
| **SciBERT + MTL + NER** | 69.83 | 53.81 | 84.06 | 70.11 | 87.89 | 75.17 |
| **SciBERT + MTL + NER + KD + Both** | 66.42 | 50.46 | 86.37 | 71.58 | 89.76 | 75.94 |
| **SciBERT + MTL + NER + KD + Both + Cosine** | 65.42 | 48.53 | 87.03 | 69.87 | 90.43 | 76.39 |
| **SciBERT + MTL + MWEs** | 62.54 | 50.86 | 82.95 | 68.13 | 90.78 | 75.95 |
| **SciBERT + MTL + MWEs + KD + Both** | 65.51 | 52.63 | 83.49 | 64.32 | 90.56 | 76.22 |
| **SciBERT + MTL + MWEs + KD + Both + Cosine** | 66.93 | 52.95 | 82.98 | 65.57 | 90.84 | 76.89 |
| **SciBERT + MTL + SciTail** | 64.96 | 49.42 | 81.58 | 66.78 | 86.33 | 71.71 |
| **SciBERT + MTL + SciTail + KD + Both** | 67.18 | 51.35 | 88.09 | 69.79 | 90.77 | 75.25 |
| **SciBERT + MTL + SciTail + KD + Both + Cosine** | 69.94 | 54.87 | 87.04 | 69.09 | 89.95 | 76.03 |
| **SciBERT + MTL + ALL** | 69.04 | 52.30 | 85.71 | 68.85 | 87.30 | 72.99 |
| **SciBERT + MTL + KD + ALL + Cosine** | **75.08** | **57.08** | **89.62** | **73.09** | **90.87** | **77.30** |

Table 5: F1-score of SciBERT based multi-task models using one auxiliary task at a time in comparison with our proposed model SciBERT+MTL+ALL+KD+Cosine. 'Both' refers to using an auxiliary task and the target task.

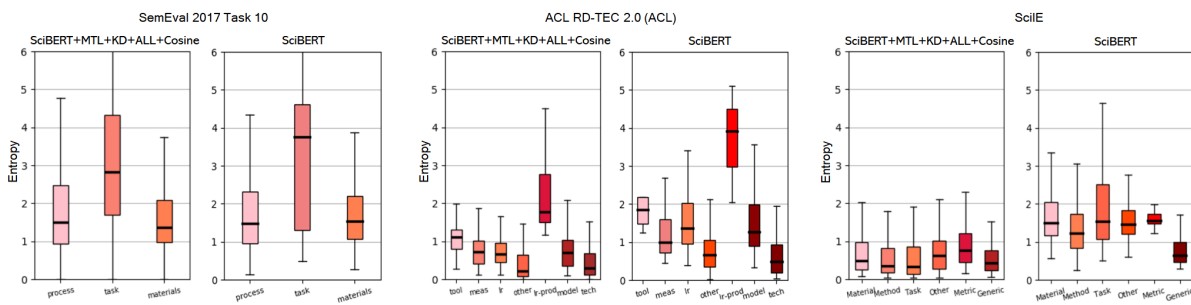

Figure 2: Entropy distribution per class between our proposed model (SciBERT+MTL+KD+ALL+Cosine) and the baseline model (SciBERT). The x-axis shows the pre-defined classes in each dataset and the y-axis shows the entropy value. The black lines indicate the median of each distribution.

We also evaluate the performance of MTL that uses all auxiliary tasks at once versus using a single auxiliary task at a time to understand the benefits of using all auxiliary tasks at once. We report the results in Table 5. We selected Chunking, NER, MWEs and SciTail as representative auxiliary tasks (to avoid clutter). We observe from Table 5 that our proposed model that combines all 11 auxiliary tasks at once performs better compared with the models that use just one auxiliary task at a time. To further investigate the performance gains on single and multiple auxiliary tasks (i.e., even subsets of the eleven auxiliary tasks), we provide the full results of MTL with single or multiple auxiliary tasks in Appendix A.3. We also find that MTL with single or multiple auxiliary tasks (but less than 11) yield lower performance compared with our proposed model on all datasets. This suggests that all auxiliary tasks are necessary to achieve the best performance. We also investigate the role of auxiliary tasks' dataset sizes in our proposed model by performing an experiment on varying the amounts of data on all auxiliary tasks. We show these results in Appendix A.4. We find that the larger the auxiliary task dataset sizes, the better the performance.

## 6.4 Analysis

### 6.4.1 Entropy Analysis

To gain more insight into our proposed model's prediction behavior in identifying and classifying scientific keyphrases, we investigate the entropy of the prediction decisions made by our model SciBERT+MTL+KD+ALL+Cosine and contrast it with the vanilla SciBERT. Entropy is a standard measurement to determine uncertainty of a probability distribution. Given a random variables $X$ with $n$ outcomes $x_1, x_2, \ldots, x_n$, the entropy of $X$ is defined as $H(X) = -\sum_{i=1}^{n} P(x_i) log P(x_i)$ (Brown et al., 1992). High entropy means high uncertainty.

We aim to understand how certain our models are in classifying keyphrases, i.e., how different behaviors emerge in models' output distribution. We do so by using entropy to explore prediction uncertainty. Figure 2 shows a histogram of entropy of

| Article from SemEval 2017 Task 10 |
|---|
| In this paper a comparison between two popular feature extraction methods is presented. Scale-invariant feature transform (or SIFT) is the first method. The Speeded up robust features (or SURF) is presented as second. These two methods are tested on set of depth maps. Ten defined gestures of left hand are in these depth maps. The Microsoft Kinect camera is used for capturing the images [1]. The Support vector machine (or SVM) is used as classification method. The results are accuracy of SVM prediction on selected images. |
| **Gold Keyphrases** |
| ('Scale-invariant feature transform', 'Process') ('SIFT', 'Process') ('Speeded up robust features', 'Process') ('SURF', 'Process') ('depth maps', 'Material') ('Support vector machine', 'Process') ('SVM', 'Process') ('comparison between two popular feature extraction methods', 'Task') ('Microsoft Kinect camera', 'Process') ('classification method', 'Process') |
| **Keyphrase outputs of SciBERT+MTL+KD+ALL+Cosine** |
| ('Scale-invariant feature transform', 'Process') ('SIFT', 'Process') ('SURF', 'Process') ('depth maps', 'Material'), ('Support vector machine', 'Process') ('SVM prediction', 'Process') ('comparison between two popular', 'Task') ('feature extraction methods', 'Process') ('Microsoft Kinect camera', 'Process'), ('defined gestures', 'Process') ('capturing', 'Process'), ('images', 'Material') ('classification method', 'Process') |

Table 6: The comparison between gold and predicted keyphrases of our model on a paper from SemEval 2017. Underlined Texts mark error predictions.

each model's output distribution shown per class in each dataset. Most notably, our proposed model has much lower entropy values in predicted keyphrases across all the classes in each dataset. For example, on SciIE, our proposed model's entropies are aligned in a similar range of low values, whereas the baseline model produces a broader range of high entropy values. We also observe similar patterns in the other two datasets. This result suggests that our proposed model is not only more accurate than the baseline model but also it is more certain in the predictions made.

### 6.4.2 Error Analysis

We manually investigate test errors to understand potential drawbacks of our proposed model. Table 6 presents an example article from the SemEval dataset and the comparison between gold labels and our proposed model's predicted keyphrases. As shown in the table, our proposed model fails to identify and classify the longest keyphrase in the article, which is '*comparison between two popular feature extraction*' annotated as *Task*. Our proposed model returns two different keyphrases which are '*comparison between two popular*' as *Task* and '*feature extraction methods*' as *Process*. It is reasonable to infer '*feature extraction methods*' as *Process*. The model has also trouble predicting whether the output is *Task* or *Process* because there is a subjectivity between the two classes. Similar types of errors can be observed in other documents of the dataset and across the datasets. This type of error is not necessarily a shortcoming of our model, but rather of the subjectivity of human annotations.

## 7 Conclusion

In this paper, we addressed the task of keyphrase boundary classification using multi-task knowledge distillation. We further enhance this approach by proposing a novel extension in which the student model not only imitates teachers' output distribution over classes but also produces similar language representations. The results of experiments on three datasets showed that our model achieves state-of-the-art results. In the future, we plan to expand our method to various settings such as few-shot learning and semi-supervised settings.

## 8 Limitations

We proposed knowledge distillation with embedding constraints for keyphrase boundary classification in which the student (multi-task model) not only learns from the teachers (single-task models) output distributions but also learns hidden language representations that do not diverge from those of the teachers. While we achieved competitive performance compared to strong baselines, there is one limitation of our method in which our proposed method struggles in classifying longer keyphrases.

## Acknowledgements

This research is supported in part by NSF CAREER award 1802358, NSF IIS award 2107518, and UIC Discovery Partners Institute (DPI) award. Any opinions, findings, and conclusions expressed here are those of the authors and do not necessarily reflect the views of NSF. We also thank our anonymous reviewers for their constructive feedback and comments, which helped improve our paper.

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

## A  Suuplementary Materials

### A.1  Auxiliary Tasks

In this section, we provide details of our auxiliary tasks by giving a specific example from each task.

**Supersense**  Supersense tagging (Johannsen et al., 2014) is the task of assigning high-level ontological classes to open-class words using Semcor 3.0 corpus.[3] In this work, we use the most common top 3 noun supersense classes which are *Person, Location* and *Group*. For example, in a given sentence '*Clara Harris, one of the guests in the box, stood up and demanded water*', the supersense of '*Clara Harris*' is *Person* and the supersense of '*guests*' is *Person*.

**MWEs**  Multiword Expressions Identification and Classification (MWEs) is the task of detecting and classifying the span of single and multiword noun and verb expressions to high-level ontological semantic classes using Streusle corpus[4] (Schneider and Wooters, 2017). In this task, we focus on both noun and verb high-level semantic classes. For example, in a given sentence '*I offer compassionate approachable and personalized counseling services*', the high-level ontological semantic class of '*compassionate*' is *verb.social*, '*personalized*' is *verb.social*, and '*counseling services*' is *noun.act*.

**FrameNet**  FrameNet Target Identification (Das et al., 2014) is the task of deciding which word span evoke a semantic frame in a given sentence of FrameNet 1.7 (Schneider and Wooters, 2017). FrameNet is a corpus of lexical and predicate-argument semantics in English. In this task, a frame is a conceptual structure describing events, relations, objects and the participants in it. For example, the target '*moist*' in a sentence evokes frame *Being* state.

**Chunking**  Text Chunking is the task of detecting the chunks of words in CoNLL-2000 shared dataset (Tjong Kim Sang and Buchholz, 2000). The chunk tags include various grammatical classes such as noun phrase, verb phrase, prepositional phrase and these tags follow the BIO schema. For example, a sentence '*He reckons the current account deficit will narrow to only #1.8 billion in September*' can

---

[3]https://web.eecs.umich.edu/ mihalcea/downloads.html
[4]https://github.com/nert-nlp/streusle

be annotated as, '[NP He] [VP reckons] [NP the current account deficit] [VP will narrow] [PP to] [NP only # 1.8 billion] [PP in] [NP September]', where NP refers to noun phrase, VP refers to verb phrase, PP refers to prepositional phrase.

**POS Tagging**   POS Tagging is the task of labeling each word in a sentence with its part of speech tag. In this work, we employ the English POS tagging annotation collection of universal dependency parsing (McDonald et al., 2013). For example, the following sentence, *'Aesthetic appreciation and spanish art.'*, has the grammar class sequence as *['ADJ', 'NOUN', 'CCONJ', 'ADJ', 'NOUN'].*

**NER**   Named Entity Recognition is the task of classifying named entities that are present in a text. The entity tags include *Person, Organization* and *Place* (Tjong Kim Sang and De Meulder, 2003).

**SciTail**   SciTail (Khot et al., 2018) is the task of recognizing the entailment of a hypothesis that is constructed from a science question and its corresponding answer by employing the premise. The dataset is collected by crowd-sourcing and by multiple-choice science questions from 4th-grade and 8th-grade exams. For example, the relation between the following two sentences, *'Neurons receive information from dendrites which are then passed to the soma cell body.'* and *'Dendrites from the cell body receives impulses from other neurons.'* is labeled as *Entail*.

**WNLI**   The Winograd Schema Challenge (Levesque et al., 2012) is the reading comprehension task where a system read a sentence with a pronoun and select the proper pronoun referent from a list of choices. Wang et al. (2019) transform this challenge as a inference task that predicts if the sentence with the pronoun substituted is entailed by the original sentence.

**RTE**   The Recognizing Textual Entailment (RTE) is the task of recognizing entailment between two sentences. The dataset is collected from a series of annual textual entailment challenges (Wang et al., 2019).

**MRPC**   The Microsoft Research Paraphrase Corpus (Dolan and Brockett, 2005) is the corpus aiming to determine whether sentences in a pair are semantically equivalent or paraphrasing one another.

**CoLA**   The Corpus of Linguistic Acceptability (Warstadt et al., 2019) is the corpus targeting to confirm grammatical acceptability judgements in a given sentence. The corpus are drawn from books and journal articles on linguistic theory.

## A.2   Training Details

Our implementation is based on the Huggingface implementation of BERT (bert-base-uncased) and SciBERT (scibert-scivocab-uncased). All the texts are chopped to span no longer than 512 tokens. We train all models using the SGD optimizer. We estimate hyper-parameters for each deep neural model via a grid search over combinations. We use the following range of values to determine the best hyper-parameters: batch size {1,4,8,16,32,64}, learning rate [0.01, 0.005], and momentum [0.0, 0.9]. The information on the best hyper-parameters setting is provided as follows. For all multi-task learning models including knowledge distillation based models, and vanilla BERT, we use momentum as 0.0, whereas vanilla SciBERT uses momentum 0.9. The training is stopped once convergence is determined on the validation dataset with 10 epochs. For each target dataset, we use batch size 1. The auxiliary tasks of Supersense, MWEs, FrameNet use batch size 4, whereas Chunking uses 32, POS Tagging uses 64, NER uses 32, SciTail uses 16, WNLI uses 16, RTE uses 1, MRPC uses 16, CoLA uses 16. Aside from these details, we follow the SciBERT paper for all other training hyper-parameters. We set $\alpha = 1$ to calculate a final training objective. All experiments are done on an NVIDIA V100 GPU.

## A.3   Effect of Auxiliary Tasks

To explore the effect of auxiliary tasks in our method, we investigate our proposed model by using a single or a subset of auxiliary tasks (i.e., less than eleven auxiliary tasks). Accordingly, we aim to: (1) understand the performance gains on models varying the number of auxiliary tasks and (2) explore the necessity of using all auxiliary tasks in our proposed model.

We first employ auxiliary tasks one at a time on SciBERT+MTL+ KD+*+Cosine (with * being a single auxiliary task) to find which auxiliary task contributes more to improve performance and show the results in the left two columns of Table 7 under the heading "Single Aux Task". We sort auxiliary tasks in decreasing order by classification tasks' F1-score. Interestingly, we observe that Chunking improves the most in perfor-

| | Single Aux Task | | Adding Aux Task Sequentially | |
|---|---|---|---|---|
| | **SemEval 2017 Task 10** | | | |
| | *Identif* | *Classif* | *Identif* | *Classif* |
| Chunking | 72.12 | 54.93 | 72.12 | 54.93 |
| SciTail | 69.94 | 54.87 | 68.77 | 52.09 |
| Supersense | 72.62 | 54.43 | 71.12 | 54.15 |
| MRPC | 66.36 | 53.64 | 60.91 | 48.65 |
| MWEs | 66.93 | 52.95 | 63.91 | 49.46 |
| CoLA | 68.08 | 52.23 | 69.28 | 53.72 |
| FrameNet | 67.30 | 52.12 | 68.39 | 53.02 |
| WNLI | 67.66 | 51.02 | 68.94 | 54.26 |
| POS Tagging | 66.75 | 50.92 | 72.11 | 55.79 |
| NER | 65.42 | 48.53 | 73.97 | 56.38 |
| RTE | 61.77 | 47.25 | 75.08 | 57.08 |
| | **ACL RD-TEC 2.0** | | | |
| | *Identifi* | *Classif* | *Identifi* | *Classif* |
| Chunking | 87.92 | 70.23 | 87.92 | 70.23 |
| NER | 87.03 | 69.87 | 87.43 | 72.38 |
| CoLA | 87.67 | 69.44 | 85.54 | 68.57 |
| Supersense | 88.85 | 69.17 | 85.04 | 67.42 |
| SciTail | 87.04 | 69.09 | 81.83 | 64.11 |
| WNLI | 88.72 | 68.18 | 86.49 | 67.72 |
| RTE | 87.58 | 67.47 | 85.38 | 65.37 |
| POS Tagging | 87.78 | 67.28 | 86.82 | 68.86 |
| MRPC | 86.35 | 66.57 | 87.47 | 70.44 |
| MWEs | 82.98 | 65.57 | 88.41 | 72.63 |
| FrameNet | 85.41 | 64.61 | 89.62 | 73.09 |
| | **SciIE** | | | |
| | *Identifi* | *Classifi* | *Identifi* | *Classifi* |
| MWEs | 90.84 | 76.89 | 90.84 | 76.89 |
| NER | 90.43 | 76.39 | 90.66 | 76.57 |
| SciTail | 89.95 | 76.03 | 88.80 | 74.65 |
| WNLI | 89.75 | 75.72 | 88.38 | 73.01 |
| Supersense | 90.53 | 75.37 | 90.63 | 76.35 |
| CoLA | 88.47 | 73.96 | 89.35 | 75.56 |
| MRPC | 87.96 | 73.32 | 89.72 | 74.95 |
| RTE | 88.20 | 72.87 | 86.89 | 72.33 |
| POS Tagging | 89.44 | 72.59 | 90.78 | 76.14 |
| FrameNet | 87.70 | 71.94 | 90.10 | 73.65 |
| Chunking | 86.93 | 71.46 | 90.87 | 77.30 |

Table 7: F1-score of SciBERT+MTL+KD+*+Cosine models where * refers to a single auxiliary task or subset of them.

| **SemEval 2017 Task 10** | | |
|---|---|---|
| | *Identifi* | *Classifi* |
| Chunking | 72.12 | 54.93 |
| Chunking + POS | 72.38 | 56.23 |
| Chunking + POS + NER | 72.41 | 55.16 |
| Chunking + POS + NER + RTE | 73.45 | 56.77 |
| **ACL RD-TEC 2.0** | | |
| Chunking | 87.92 | 70.23 |
| Chunking + NER | 86.91 | 72.31 |
| Chunking + NER + MRPC | 86.42 | 71.02 |
| Chunking + NER + MRPC + MWEs | 86.54 | 71.53 |
| Chunking + NER + MRPC + MWEs + FrameNet | 88.11 | 72.54 |
| **SciIE** | | |
| MWEs | 90.84 | 76.89 |
| MWEs + NER | 89.05 | 76.31 |
| MWEs + NER + Supersense | 90.23 | 76.39 |
| MWEs + NER + Supersense + POS | 90.22 | 76.61 |
| MWEs + NER + Supersense + POS + Chunking | 90.58 | 76.87 |

Table 8: F1-score of SciBERT+MTL+KD+*+Cosine, where * is a set of sorted auxiliary tasks that increase the target task performance in the initial and the final phase on multi-task learning of sequentially adding auxiliary tasks.

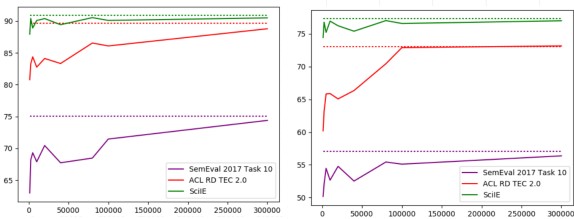

Figure 3: F1-score on the impact of auxiliary tasks data size on the target task keyphrase boundary identification(left)/classification(right) performance in SciBERT+MTL+KD+ALL+Cosine. Dotted lines refer to the best performance of SciBERT+MTL+KD+ALL+Cosine model on each dataset.

mance on SemEval and ACL, whereas Chunking improves the least in performance on SciIE. To further investigate the impacts of the auxiliary tasks, we sequentially add auxiliary tasks one at a time on SciBERT+MTL+KD+*+Cosine model following the order and show results in the right two columns of Table 7 under the heading "Adding Aux Task Sequentially". We observe that the performance generally increases in the initial and the final phase of adding sorted auxiliary tasks on the SciBERT+MTL+KD+*+Cosine (with * being a set of auxiliary tasks) model. Hence, we additionally run a set of experiments using only a few auxiliary tasks selected from the top and the end of the ranked list. The results are reported in Table 8. We find that combining initial and final ranked

auxiliary tasks generally achieve competitive performance. However, none of the models achieve performance as good as our proposed model which uses all auxiliary tasks at once. This supports that all auxiliary tasks are required to achieve competitive performance for the keyphrase boundary classification task.

### A.4 Effect of Auxiliary Tasks Dataset Size

We investigate the impact of dataset size of the auxiliary tasks on our SciBERT+MTL+KD+ALL+Cosine model's performance. Specifically, we conduct a set of experiments on varying amounts of eleven auxiliary tasks' data size on the proposed model and show the results in Figure 3. Interestingly, we observe that our proposed model obtains the largest performance improvements when using a large amount of auxiliary tasks' data size even though we use a small amount of target task

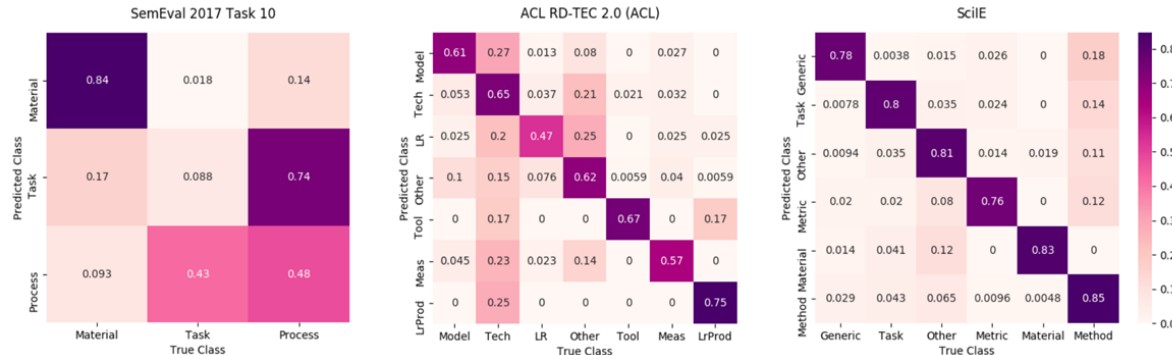

Figure 4: The keyphrase classification results - confusion matrix visualization of each target dataset. The x-axis refers to true classes and the y-axis refers to predicted classes in each target task. The numbers are normalized by the count of predicted keyphrases

training samples.

## A.5 Error Analysis

We visualize confusion matrices of our proposed model output (SciBERT+MTL+ALL+KD+Cosine) in Figure 4. In Figure 4, numbers of each cell represent how many classified keyphrases belong to each true class. For example, on SemEval 2017 Task 10 dataset confusion matrix, the cell corresponding to row *Process* and column *Task* refers to the ratio of keyphrases predicted as *Task* but which should be classified as *Process* to the total number of keyphrases that are classified as *Task*. Consequently, each row in every confusion matrix sums up to 1. We also confirm the subjectivity of pre-defined classes poses a challenge to keyphrase boundary classification task. For the SemEval 2017 Task 10 dataset, we observe our proposed model incorrectly classifies keyphrases between *Task* and *Process* because there is subjectivity between these two classes. For the ACL RD-TEC 2.0 dataset, our proposed model mis-classifies keyphrases as *Tech*. However, in contrast to the above two target datasets, on the SciIE dataset, we observe that our proposed model generally performs well.