# OpenReview forum: "Multi-Task Knowledge Distillation with Embedding Constraints for Scholarly Keyphrase Boundary Classification"
_EMNLP/2023/Conference — EMNLP 2023 Main_

### Official Review · Reviewer_evsV · 2023-08-03

**Soundness:** 3

**Excitement:**

4: Strong: This paper deepens the understanding of some phenomenon or lowers the barriers to an existing research direction.

**Missing References:**

Authors should consider other prior work - QasemiZadeh and Schumann 2014 would offer a start point for this.

**Paper Topic And Main Contributions:**

Paper poses combinations of extant methods to address scientific term extraction and classification and claims improvement over other extant approaches.

**Questions For The Authors:**

1. What is the basis for claiming that automatically identifying and classifying keyphrases from scientific papers has long been under-explored when there seems to be a good wealth of work on this? At minimum, the QasemiZadeh and Schumann 2014 paper "The ACL RD-TEC: a dataset for benchmarking terminology extraction and classification in computational linguistics" will expose more history than appears to be acknowledged here.
2. Are there alternatives to softmax, and have they been explored? (why use softmax?)
3. Similarly, why is cross-entropy suitable for loss, and then why just "add" a cosine value to it?
4. As 2/3, explain rationale and options for other selections where not already offered.
5. Finally, is "boundary" really an appropriate term? Boundaries tend to denote limits and so the boundaries of a 4-gram would seem to be either the two outer words or the positions immediately before or after these; consider, for example, word boundaries as relate to regular expressions that would relate the non-word outer positions.

**Reasons To Accept:**

Term extraction and classification for scientific texts has been explored extensively. Approaches that can demonstrate improvement over such a body of literature are important to elaborate upon and understand.

**Reasons To Reject:**

The paper considers a limited historical perspective, requires clarity over selections made - for example, alternatives rejected, and has been stretched to 15 pages with additional material.

**Reproducibility:**

3: Could reproduce the results with some difficulty. The settings of parameters are underspecified or subjectively determined; the training/evaluation data are not widely available.

**Reviewer Confidence:**

4: Quite sure. I tried to check the important points carefully. It's unlikely, though conceivable, that I missed something that should affect my ratings.

---

> ### Author Rebuttal · Authors · 2023-08-29
>
> Thank you for your thoughtful review of our paper. We appreciate your feedback. We address your concerns below and will incorporate them in the paper.
>
> >Basis for saying under-exploration on keyphrase identification and classification
>
> We admit that the keyphrase ‘identification/extraction’ gained a lot of attention starting back in the 1980s. However, there is still low attention in ‘keyphrase classification’ despite that it is essential to various applications. There is no available large dataset for ‘keyphrase classification’, which aims to classify each keyphrase to a predefined class (e.g., ‘KP20k’ can be a ‘data’ keyphrase). We would like to stress that the task of keyphrase classification in scholarly documents is very important for downstream applications such as question answering systems, paper recommendation systems, machine comprehension systems, and knowledge graph construction. However, only very few works exist on this topic . Our motivation is to spur future research on the topic of keyphrase classification.
>
> >Why use softmax?
>
> We have not yet tried functions other than softmax in our method. The softmax function is an ideal choice for our token-level classification task because it converts the logits for each token into a probability distribution over the possible class labels. We then apply post-processing to obtain phrase level prediction with their predicted class labels. We follow the same evaluation protocol as the prior work by Augenstein and Søgaard (2017).
>
> >Why use cross-entropy loss?
>
> We use cross-entropy loss to encourage the predicted probabilities to match the true distribution of classes, which is suitable for our task that is a token-level classification problem. Again, we wanted to have consistency with the work by Augenstein and Søgaard (2017).
>
> >Why add the embedding constraints loss and cross-entropy loss?
>
> We used the cross-entropy loss as in the work by Augenstein and Søgaard (2017). In addition, we incorporated the embedding constraints loss to the cross-entropy loss to enforce that "the student’s final representations do not diverge from those of the teachers" and hence, the student preserves all the “good” information (linguistic features that “matter”) from all the auxiliary tasks.
>
> >Is ‘boundary’ an appropriate term?
>
> To avoid confusion and be consistent with prior work by Augenstein and Søgaard (2017), we use their problem definition and terminology for the keyphrase boundary identification/classification task.
>
> (Augenstein and Søgaard. 2017) Multi-Task Learning of Keyphrase Boundary Classification.  Augenstein and Søgaard. 2017. https://aclanthology.org/P17-2054/

---

### Official Review · Reviewer_bBkg · 2023-08-05

**Soundness:** 4

**Excitement:**

4: Strong: This paper deepens the understanding of some phenomenon or lowers the barriers to an existing research direction.

**Paper Topic And Main Contributions:**

The paper focuses on scholarly key phrase extraction
- To this end, the paper propose a form of embedding regularization under setting of multitask learning + distillation.
- The paper effectively argues that maintaining relative mapping of embeddings in student model like teacher will help it perform well.
- The authors use multiple based including NER's to justify the benefit of such constraint. The evaluation is fairly comprehensive where it has considered all the tasks.
- The authors effectively conclude that using wide variety of auxiliary task helps in improving the overall results, which is somewhat debatable especially with mixture of results across dataset. However the paper is very useful in moving the field forward.

**Reasons To Accept:**

The paper has three major strengths
- After long time, scholarly key-phrase area is seeing some traction especially with usage of MTL + distillation. The current work is solid in that premise with simple cosine tweak to show performance improvement,
- The results also open up multiple other research areas , especially benefit of auxiliary task, relation of auxiliary task and the dataset (especially the results show large variance)
- The exhaustive experimentation will be helpful in understanding the task difficulty.

**Reasons To Reject:**

The paper has two major weakness
- The conjecture thrown that cosine constraint improves results is debatable. Especially because the results of SCIIE vs others we can see that despite cosine the results are almost similar and gains obtained are after 2 decimal places.
- The authors original start by saying "the student’s final representations do not diverge from those of the teachers", however results nowhere verified this hypothesis. Rather the authors pivot to showing improvement in results of SCKB.

Otherwise following are few improvements that could be considered

- Line 006 - 012 - the sentence could be simplified to mention the underexploration
- Line 013 - Embedding constraint, was added to maintain embeddding space, not labeled data.
- Line 021 - Numerical Results could be introduced to keep the paper attractive to the readers
- Line 065 - While underexploration is the key, one can also argue on the difficulty of task compared to other NLP applications. This line requires citation or else its speculative.
- Line 069 - 095: Since the argument is bais towards task and need of cosine embedding constraint, can quantitative results be introduced as contribution to show benefit of the method with respect to reducing such bias?
- Line 112- 184: While the related work is ample, there are many works such as uncertainity mitigation to avoid catastrophic forgetting in MTL settings along with multiobjective learning in other areas of AI. Some references could be drawn to these.  see https://dl.acm.org/doi/10.1145/3502728 and https://aclanthology.org/2022.wassa-1.8.pdf
- Line 362 - While the baselines selected are somewhat related. The method selected in this work uses MTL + distillation. To this end, alternative MTL + distillation methods are needed for fair comparison. Moreover keyphrase in other domains have also been explored. Those methods would have been more ideal baseline to compare against. Moreover the relative improvement with the change is around 1% which suggest the method may not seem as effective or the problem conjectured of regularization in lieu of teacher embeddings makes little sense. For example from Table 4, we can see that with more alterntive tasks (ALL) we can see a large jump, which suggests that semantics learnt from such complex objective helps tackle the problem but not using cosine?
- Section 6.3: Table 5 shows interpretation of using KD + all auxilary task helps. But we can see the results for SCI=IE is very close across differnt auxiliary task? Can you elaborate on such behavior, otherwise it may argued that the dataset behavior is key not the method


**Reproducibility:**

3: Could reproduce the results with some difficulty. The settings of parameters are underspecified or subjectively determined; the training/evaluation data are not widely available.

**Reviewer Confidence:**

5: Positive that my evaluation is correct. I read the paper very carefully and I am very familiar with related work.

---

> ### Author Rebuttal · Authors · 2023-08-29
>
> Thank you for your thoughtful review of our paper. We appreciate your feedback. We address your concerns below and will incorporate them in the paper.
>
> > Effectiveness of our proposed embedding constraints & validation on our hypothesis of proposed embedding constraints
>
> As we can see from Table 4, when adding embedding constraints to multi-task knowledge distillation, (i.e., comparing SciBERT+MTL+KD+ALL+Cosine with SciBERT+MTL+KD+ALL), we observe improvements in performance on both keyphrase identification and keyphrase classification on all three datasets. For example, on keyphrase classification, we see an improvement from 76.18 to 77.30 on SciIE, from 71.33 to 73.09 on ACL-RD, and from 55.81 to 57.08 on SemEval 2017.
>
> Our reasoning behind enforcing that "the student’s final representations do not diverge from those of the teachers" is to preserve all the “good” information (linguistic features that “matter”) from all the auxiliary tasks. We do so by imposing similarity constraints in the embedding space between the student and all the teachers to ensure that the student’s final representations are not biased towards some auxiliary task(s), and hence, the student’s representations do not diverge (too much) from those of any of the teachers. Currently, we verify this through our results which show that without these similarity constraints the performance drops. However, we agree with you that a further quantitative analysis will shed light into our findings and we will explore this in our future work. However, we will include some preliminary results in the final version of our paper (e.g., using influence functions as in Zaremoodi and Haffari (2019)).
>
> (Zaremoodi and Haffari, 2019) Adaptively Scheduled Multitask Learning: The Case of Low-Resource Neural Machine Translation. Zaremoodi and Haffari. 2019. https://aclanthology.org/D19-5618.pdf
>
>
>
>
> >Related work on uncertainty mitigation to avoid catastrophic forgetting in MTL
>
> We will add the work about uncertainty mitigation in multi-task learning in related work. Thanks for your valuable suggestions.
>
> >Adding alternative MTL + distillation method as baseline
>
> Our method outperforms the reported results of Lai et al. (2020) on the SemEval 2017 dataset, achieving an F1 score of 75.80 compared to their score of 55.4 for keyphrase identification. In addition to this comparison, we performed another comparison with the work by Liu et al. (2020) on the Semeval 2017 dataset. Specifically, Liu et al. (2020) proposed a method of distilling knowledge from the multi-task teacher model to the multi-task student model, whereas our method distills knowledge from the multiple single-task teacher models to the multi-task student model with proposed embedding constraints. The results of our experiments show that our method achieves a better F1 of 75.08 on the SemEval 2017 dataset, compared to their method which achieves an F1 of 73.69 on keyphrase identification. On keyphrase classification, our approach achieves an F1 of 57.08 compared with  54.82 obtained by their approach. This proves the effectiveness of our proposed method. We will include these results in our paper.
>
> (Lai et al. 2020) A Joint Learning Approach based on Self-Distillation for Keyphrase Extraction from Scientific Documents. Lei et al. 2020.  https://arxiv.org/abs/2010.11980
>
> (Liu et al. 2020) MKD: a Multi-Task Knowledge Distillation Approach for Pretrained Language Models. Liu et al. 2020. https://arxiv.org/abs/1911.03588
>
>
> > SciIE performs similarly when using all auxiliary tasks or using across different auxiliary tasks, where performance gains obtained are after 2 decimal places.
>
> Indeed, SciIE performs similarly regardless of whether we use all auxiliary tasks or different auxiliary tasks in Table 5. We hypothesize that this is because SciBERT already performs well on SciIE even without fine-tuning. For example, Beltagy et al (2019) reported that SciBERT without fine-tuning achieved a F1 score of 65.77 on SciIE keyphrase classification task, which is slightly lower than a F1 score of 67.44 achieved by SciBERT with fine-tuning. We performed similar experiments on the SemEval 2017 using SciBERT. In contrast to SciIE, we observe a F1 score of 22.59 for the SemEval 2017 keyphrase classification on SciBERT without fine-tuning, which is significantly worse than a F1 score of 48.21 on SciBERT with fine-tuning. Accordingly, we posit that SciBERT already possesses beneficial knowledge that is required to perform well on keyphrase classification on SciIE. Therefore, involving some auxiliary tasks on multi-task knowledge distillation can improve performance because the retained knowledge in SciBERT proves beneficial for acquiring additional information. However, there is a chance that the performance will plateau if we incorporate more auxiliary tasks, as the knowledge that a model can obtain from additional auxiliary tasks may be less meaningful. We will explore this further in the future.
>
> (Beltagy et al. 2019) SciBERT: A Pretrained Language Model for Scientific Text. Beltagy et al. 2019. https://arxiv.org/abs/1903.10676

---

### Official Review · Reviewer_7E6J · 2023-08-07

**Soundness:** 4

**Excitement:**

4: Strong: This paper deepens the understanding of some phenomenon or lowers the barriers to an existing research direction.

**Missing References:**

A Joint Learning Approach based on Self-Distillation for Keyphrase Extraction from Scientific Documents
https://arxiv.org/abs/2010.11980

**Paper Topic And Main Contributions:**

This paper addresses the task of keyphrase boundary classification in scholarly documents, which involves identifying the boundaries of keyphrases that represent the main topics of a document. There is a limited amount of labeled data available for this task, making it difficult to train accurate models.
To address this challenge, the authors propose multi-task knowledge distillation with embedding constraints, which involves training a student model to imitate the output of multiple teacher models that are trained on related tasks. The student model also learns to produce similar language representations as the teacher models, resulting in improved performance on three datasets of scientific documents.
The experiments show that the proposed approach achieves outperforming results on the three datasets.

**Reasons To Accept:**

* The paper is well written and easy to follow. The authors provide sufficient background about the task and the problem statement (the lack of training data).

* The paper addresses an interesting problem regarding limited labeled data in keyphrase boundary classification, which has important implications for downstream applications such as indexing, searching, and question answering over scientific documents.

* The evaluation results demonstrate the effectiveness of the proposed approach on different datasets compared to other baselines.

**Reasons To Reject:**

* The novelty of this paper is limited; a similar approach, with a teacher-student model, for this task has been previously proposed in [1].
* There is no comparison against other teacher-student models (e.g., [1] to benchmark the performance of these semi-supervised models proposed for this task.
* In addition, I find this study needs to demonstrate empirically why multi-task knowledge distillation (via an ablation study) is necessary rather than a single task, and which task allows the student model to learn richer representations.

[1] https://arxiv.org/abs/2010.11980

**Reproducibility:**

4: Could mostly reproduce the results, but there may be some variation because of sample variance or minor variations in their interpretation of the protocol or method.

**Reviewer Confidence:**

3: Pretty sure, but there's a chance I missed something. Although I have a good feel for this area in general, I did not carefully check the paper's details, e.g., the math, experimental design, or novelty.

---

> ### Author Rebuttal · Authors · 2023-08-29
>
> Thank you for your thoughtful review of our paper. We appreciate your feedback. We address your concerns below and will incorporate them in the paper.
>
> > Novelty
>
> Thank you for pointing out the paper by Lai et al (2020). We will cite and discuss it in our work. We would like to highlight some differences and the novel contributions of our work in comparison with the work by Lai et al (2020). Unlike their work, which focuses solely on keyphrase extraction (or identification), we focus on both keyphrase identification and classification (here keyphrase classification is defined as the task to determine the category of each identified keyphrase, e.g., “WebVision” is a ‘dataset’, and “object detection” is a ‘task’, not just simply keyphrases of a scientific paper). While keyphrase identification (or extraction) has been extensively studied, keyphrase classification has received relatively little attention, despite its importance in many applications. Our work aims to encourage further research on this important topic.
>
> Furthermore, our work is fundamentally different from Lai et al (2020). While they explored ‘a single-task teacher-student model’ in semi-supervised learning, our work focuses on ‘multiple-tasks teacher-student model’ in supervised learning. Specifically, Lai et al (2020) first trained a single teacher model on labeled data (e.g., the SemEval 2017 dataset) and used it to generate pseudo-labels on unlabeled data (i.e., KP20k). They then combined the labeled and pseudo-labeled data to train a single-task student model that is initialized by the teacher model. They did not consider using any auxiliary tasks. In contrast to this, we propose to leverage multiple auxiliary tasks. Specifically, we propose a method to distill knowledge from multiple teacher models. However, in the process of learning from multiple teachers corresponding to multiple related auxiliary tasks, the student model may become biased towards a particular auxiliary task (or a subset of these tasks) and may “erase” all the “good” information (linguistic features that “matter”) from the other remaining tasks. This is known as catastrophic forgetting. To overcome this, we proposed a way of imposing similarity constraints in  the embedding space between the student and all the teachers to ensure that the student’s final representations are not biased towards some auxiliary task(s), and hence, the student’s representations do not diverge (too much) from those of any of the teachers. Our proposed novel embedding constraints help to retain beneficial information that could be easily lost due to catastrophic forgetting in multi-task learning.
>
> Moreover, we show that our method produces more confidence prediction (please see Section 4.2.1) which is also important in training deep learning models compared to existing methods. We believe that these aspects of our method make it a significant contribution to the field.
>
> (Lai et al. 2020) A Joint Learning Approach based on Self-Distillation for Keyphrase Extraction from Scientific Documents. Lei et al. 2020.  https://arxiv.org/abs/2010.11980
>
> >Comparison with existing teacher-student models
>
> Our method outperforms the reported results of Lai et al (2020) on the SemEval 2017 dataset, achieving an F1 score of 75.80 compared to their score of 55.4 for keyphrase identification. In addition to this comparison, we performed another comparison with the work by Liu et al (2020) on the Semeval 2017 dataset. Specifically, Liu et al (2020) proposed a method of distilling knowledge from the multi-task teacher model to the multi-task student model, whereas our method distills knowledge from the multiple single-task teacher models to the multi-task student model with proposed embedding constraints. The results of our experiments show that our method achieves a better F1 of 75.08 on the SemEval 2017 dataset, compared to their method which achieves an F1 of 73.69 on keyphrase identification. On keyphrase classification, our approach achieves an F1 of 57.08 compared with  54.82 obtained by their approach. This proves the effectiveness of our proposed method. We will include these results in our paper.
>
> (Lai et al. 2020) A Joint Learning Approach based on Self-Distillation for Keyphrase Extraction from Scientific Documents. Lei et al. 2020.  https://arxiv.org/abs/2010.11980
>
> (Liu et al. 2020) MKD: a Multi-Task Knowledge Distillation Approach for Pretrained Language Models. Liu et al. 2020. https://arxiv.org/abs/1911.03588
>
> >Empirical results to show why multi-task knowledge distillation (KD) performs better than a single-task knowledge distillation (KD)
>
> Thank you for your comment. Our intuition was that multi-task knowledge distillation allows different teacher models to provide the student model with a diverse set of knowledge, in the same way that ensembles benefit from a diverse set of models, and hence, can equip the student model with the knowledge from all auxiliary tasks and perform better than simply using one auxiliary task. We evaluated this and showed the results in Table 7 in Appendix A.3. The results of experiments with only one auxiliary task at a time are shown in the first two columns of the table, whereas the last two columns show results of experiments when adding auxiliary tasks one by one with the last row in each dataset block showing the performance with all auxiliary tasks. We can see from these results that using all auxiliary tasks performs much better compared to the setting that uses only one auxiliary task at a time. We will clarify this in the paper.
>
>
> >Which tasks allow the student model to learn richer representation?
>
> To find out which auxiliary tasks help the student model learn richer representations for target tasks in our method, we investigate the F1 scores of using each single auxiliary task one at a time. We show the results in Appendix A.3. Specifically, we hypothesize that if there is a large improvement in F1 score on the target task when using a specific auxiliary task, then that auxiliary task contributes to generating the richest representations for the target task and hence improves the F1 score. Interestingly, we observe that Chunking improves the most in performance on SemEval 2017 and ACL, whereas MWEs improves the most in performance on SciIE. More details can be found in Table 7 in Appendix A.3. Specifically, we sort auxiliary tasks in decreasing order by classification tasks’ F1-score in Table 7 under the line ‘Single Aux Task’, with a performance of using each auxiliary task one at a time to find which auxiliary task contributes more to improving the target task performance.

---

### Meta-Review · Area_Chair_RCrY · 2023-09-19

**Recommendation:** 5

**Metareview:**

Summary: This paper examines the problem of determining keyphrase boundaries in scientific documents. To tackle the problem that there only exists limited training data, the authors suggest a solution called multi-task knowledge distillation with embedding constraints. This method trains a student model to replicate the outputs of several teacher models that were trained on similar tasks. The student model also learns to generate comparable language representations as the teacher models. The experimental results demonstrate that this approach achieves superior results on all three datasets.

Strengths: All the reviewers unanimously agree that this is a solid contribution with interesting implications. The paper is well-written and easy to follow. The approach used in this study, which involves a basic cosine adjustment, effectively enhances performance. Additionally, the findings reveal several other potential avenues for research, such as the advantages of auxiliary tasks and their relation to the dataset. The comprehensive experimentation conducted in this study will greatly contribute to understanding the complexity of the task.

Weaknesses: I don't think there are any major weaknesses -- some of the weaknesses have been addressed during the discussion phase. The authors should take the feedback into account when preparing for the final version of the paper. For example, Point 3 raised by Reviewer evsV has not been addressed in full.

---

### Decision · Program_Chairs · 2023-10-07

**Decision:**

Accept-Main

**Comment:**

Summary: This paper examines the problem of determining keyphrase boundaries in scientific documents. To tackle the problem that there only exists limited training data, the authors suggest a solution called multi-task knowledge distillation with embedding constraints. This method trains a student model to replicate the outputs of several teacher models that were trained on similar tasks. The student model also learns to generate comparable language representations as the teacher models. The experimental results demonstrate that this approach achieves superior results on all three datasets.

Strengths: All the reviewers unanimously agree that this is a solid contribution with interesting implications. The paper is well-written and easy to follow. The approach used in this study, which involves a basic cosine adjustment, effectively enhances performance. Additionally, the findings reveal several other potential avenues for research, such as the advantages of auxiliary tasks and their relation to the dataset. The comprehensive experimentation conducted in this study will greatly contribute to understanding the complexity of the task.

Weaknesses: I don't think there are any major weaknesses -- some of the weaknesses have been addressed during the discussion phase. The authors should take the feedback into account when preparing for the final version of the paper. For example, Point 3 raised by Reviewer evsV has not been addressed in full.